# Epigenetic and Immunological Features of Bladder Cancer

**DOI:** 10.3390/ijms24129854

**Published:** 2023-06-07

**Authors:** Irina Gilyazova, Kadriia Enikeeva, Guzel Rafikova, Evelina Kagirova, Yuliya Sharifyanova, Dilara Asadullina, Valentin Pavlov

**Affiliations:** Institute of Urology and Clinical Oncology, Bashkir State Medical University, 450008 Ufa, Russia; kienikeeva@bashgmu.ru (K.E.); rafikovaguzel@gmail.com (G.R.); emkagirova@bashgmu.ru (E.K.); yuvsharifyanova@bashgmu.ru (Y.S.); ddasadullina@bashgmu.ru (D.A.); pavlov@bashgmu.ru (V.P.)

**Keywords:** bladder cancer, epigenetic alterations, DNA methylation, miRNA, circular RNA, noninvasive detection, liquid biopsy, tumor microenvironment

## Abstract

Bladder cancer (BLCA) is one of the most common types of malignant tumors of the urogenital system in adults. Globally, the incidence of BLCA is more than 500,000 new cases worldwide annually, and every year, the number of registered cases of BLCA increases noticeably. Currently, the diagnosis of BLCA is based on cystoscopy and cytological examination of urine and additional laboratory and instrumental studies. However, cystoscopy is an invasive study, and voided urine cytology has a low level of sensitivity, so there is a clear need to develop more reliable markers and test systems for detecting the disease with high sensitivity and specificity. Human body fluids (urine, serum, and plasma) are known to contain significant amounts of tumorigenic nucleic acids, circulating immune cells and proinflammatory mediators that can serve as noninvasive biomarkers, particularly useful for early cancer detection, follow-up of patients, and personalization of their treatment. The review describes the most significant advances in epigenetics of BLCA.

## 1. Introduction

Bladder cancer (BLCA) is a prevalent malignancy affecting the urogenital system in adults. In 2020, there were 573,278 newly diagnosed cases of BLCA, resulting in 212,536 BLCA-related fatalities [1,2]. Systematic smoking is recognized as one of the primary contributors to bladder malignancy [3]. Additionally, occupational exposures and contact with carcinogens play a significant role, accounting for approximately 20–25% of BLCA cases [4]. Advanced age is a known risk factor, as over 90% of patients in the United States are above 55 years old [5]; other risk factors include Caucasian race [6,7] and the use of certain drugs containing phenacetin, cyclophosphamide, or pioglitazone [8,9,10]. The likelihood of developing urinary system neoplasms, particularly BLCA, increases in patients with chronic cystitis, bladder stones, and urinary stasis [11].

BLCA predominantly originates in the urothelium, which refers to the epithelial tissue lining the inner surface of the bladder and other urinary organs. This characteristic gives rise to urothelial carcinoma, which represents the most prevalent form of bladder cancer, constituting approximately 80–90 percent of all BLCA cases. Depending on the ability of the tumor to penetrate the muscular layer, BLCA is classified into non-muscle-invasive (NMIBC) (70% of cases of first detected tumors) and muscle-invasive BLCA (MIBC) (30%) [12]. Currently, the diagnosis of BLCA is based on cystoscopy and cytological examination of urine and additional laboratory and instrumental studies. However, cystoscopy is an invasive study, and voided urine cytology has a low level of sensitivity, so there is a clear need to develop more reliable markers and test systems for detecting the disease with high sensitivity and specificity. Despite receiving 1–3 years of maintenance therapy, patients with NMIBC face a recurrence risk exceeding 50% and a progression risk of 20% within a 5-year period [13]. Although NMIBC exhibits favorable 5-year survival rates (>90%), most patients undergo extensive cystoscopic surveillance and multiple therapeutic interventions, which negatively impact their health-related quality of life [14]. This raises the necessity for the identification of new predictive markers for the disease. Human body fluids (urine, serum, and plasma) are known to contain significant amounts of tumorigenic nucleic acids that can serve as noninvasive biomarkers, particularly useful for early cancer detection, follow-up of patients, and personalization of their treatment. Epigenetic alterations such as DNA methylation, aberrant microRNA expression, long noncoding RNAs, and circular RNA alterations can be detected in the earliest stages of cancer and are quite easily detected in biological fluids even before any clinical signs and morphological changes appear. It should be noted that BLCA is a heterogeneous disease that is difficult to treat, and it has specific genetic and epigenetic features. Early disease detection holds the potential to prevent invasion and metastasis, thereby improving overall survival outcomes for BLCA patients. However, current recommendations do not support screening for BLCA in the general population because of its limited efficacy and high economic costs [13,15]. In such circumstances, it becomes crucial to focus on the development of novel and effective biomarkers for BLCA or the refinement of high-risk groups using genetic, immune, and epigenetic markers. This approach justifies the costs associated with screening by enabling early disease detection and enhancing clinical outcomes [13]. The pursuit of these advancements holds promise for the improved management and prognosis of BLCA.

## 2. Epigenetic Alterations of BLCA

In human DNA, methylation is a molecular process that targets CpG dinucleotides. Most genes have unmethylated CpGs in their 5′-regulatory region. DNA methyltransferases (DNMTs) catalyze the methylation of these CpGs, thereby inducing silencing of transcription, leading to suppression of tumor growth and cancer cell invasion.

### 2.1. DNA Methylation

DNA methylation is a strong tool for suppressing gene activity. Hyper- and hypomethylated DNA sites are identified in BLCA and precancerous lesions. DNA methylation status can be assessed from cell-free DNA fragments and tumor cells excreted with urine. A significant predominance of methylated genes such as *APC* and *CCND2* (cyclin D2) has been detected in BLCA compared to benign cases. Hypermethylation of individual genes including *GSTP1*, *APC*, and *RARB2* has been identified in patients with urothelial BLCA [16]. 

It is important that this inverse correlation has been clearly shown only for methylation in the promoter regions but not in the coding part of the genes. Methyl groups do not have influence on base conjugation, but they can affect the protein–DNA interaction by acting in the main groove [17]. The identification of CpG residues within the promoters of certain tumor suppressor genes linked to bladder cancer (BLCA) has spurred extensive research into the role of methylation in gene suppression. Hypermethylation occurring in the promoter regions of 50 genes has been associated with transcriptional silencing, offering an alternative mechanism for suppressing gene expression in addition to gene deletion or mutation. Notably, hypermethylation of the promoter region of the *p16INK4A* gene has been observed in patients with superficial transitional cell carcinomas, suggesting its potential involvement as an early event in the pathogenesis of transitional cell carcinomas [18]. In particular, a high frequency of mutations has been found in the *KMT2D* gene (also known as *MLL2*) and in the *KDM6A* gene (also known as *UTX*), which encode histone H3 lysine 4 methyltransferase and histone lysine demethylase, respectively [19,20,21]. However, the most studied epigenetic mechanism is DNA methylation [22]. Comparing methylation patterns, Wolff et al. found global hypomethylation in NMIBC, whereas hypermethylation was more frequent in other tumors, which supports the concept that DNA methylation plays an important role in the formation and aggressiveness of BLCA, being a putative target for anticancer therapy [23].

Methylated genes, including *SFRP1*, *SOX9*, *FHIT*, *CDH1*, *PMF1*, *RUNX3*, *LAMC2*, and *RASSF1A*, cannot only be considered as biomarkers but are also associated with poor clinical outcomes in patients with BLCA [24]. This suggests the potential utility of DNA methylation as a prognostic marker. Through univariate and multivariate Cox regression analysis, genes with differential DNA methylation can be employed as prognostic biomarkers for BLCA. By utilizing this prognostic model, patients can be stratified into high- or low-risk groups. Assessing the methylation status of these genes allows for the evaluation of patient survival [25].

In urine samples from 108 UCB patients and 100 age- and gender-matched controls, the methylation levels of *FAM19A4*, *GHSR*, *MAL*, miR-129, miR-935, *PHACTR3*, *PRDM14*, *SST*, and *ZIC1* markers were quantitatively assessed using methylation-specific qPCR in urine sediment. All nine urinary methylation markers (*FAM19A4*, *GHSR*, *MAL*, miR-129, miR-935, *PHACTR3*, *PRDM14*, *SST*, and *ZIC1*) exhibited significantly higher methylation levels in BLCA patients compared to the control group (*p* < 0.001) [26]. Importantly, the methylation alterations detected in urine sediment samples resemble those observed in tumor tissues [27,28].

After performing sequential clustering to identify DNA methylation subgroups in BLCA, differences between subgroups, including DNA methylation levels, were examined. The results revealed significant prognostic differences among the four subgroups of molecular subtypes of BLCA. Thus, the study allowed for the identification of molecular subtypes of BLCA based on DNA methylation and determination of their prognostic characteristics. These findings can be valuable for clinicians in predicting patient survival and making treatment decisions [29].

Analysis of the expression of 15 DNA methylation regulators in 985 samples from six independent cohorts of BLCA revealed alterations in DNA methylation regulators at a frequency of 12.62% [30]. The *MBD1* gene had the highest frequency of alterations in BLCA, indicating its involvement in oncogenesis. The frequency of changes in *DNMT1*, *DNMT3A*, and *DNMT3B* genes was also 2%. Co-mutations were observed in several DNA methylation regulators, including *NTHL1*, *MBD3*, *MECP2*, *UHRF2*, and *ZBTB33*, despite their functional differences. 

Analysis of DNA methylation regulators revealed three different clusters (A, B, and C) based on their expression levels with different patient survival rates. Each cluster exhibited significantly different characteristics of the tumor microenvironment [30]. 

Analysis of DNA methylation profiles in the peripheral blood of 603 patients, who were part of a population cohort in New Hampshire, with NMIBC and investigating the association between methylation profiles and patient outcomes, particularly focusing on the methylation-driven neutrophil-to-lymphocyte ratio (mdNLR), revealed that increased mdNLR increases the risk of recurrence-free survival (RFS) in patients with NMIBC. For patients with high-risk recurrence of NMIBC, intravesical immunotherapy with bacillus Calmette-Guérin (BCG) is the standard treatment to induce an immune response against remaining BLCA cells after surgery, thus the immune profile of the blood represents a potential prognostic factor [31]. Thus, DNMT inhibitors are being considered as a promising therapeutic strategy for BLCA [32]. 

Some of these inhibitors, such as 5-azacytidine and decitabine, have already been approved for the treatment of specific types of blood cancer. However, these inhibitors have limitations that restrict their application. Combining them with chemotherapy or immune checkpoint inhibitors can enhance their effectiveness. Current research includes both laboratory and clinical studies, investigating the effects of DNMT inhibition on DNA methylation in the context of BLCA. This research direction presents new perspectives in the treatment of BLCA and may lead to the development of alternative or adjunctive therapeutic strategies [32].

A number of epigenetic mechanisms are disrupted in the early stages of BLCA development and progression [22]. For example, mutations in *EP300* and *CREBBP*, two chromatin remodeling genes, lead to inactivation of the histone acetyltransferase domain complex, which subsequently changes the chromatin conformation. Furthermore, gene expression associated with the loss of histone acetyltransferase activity is associated with more aggressive bladder tumors [33]. Interestingly, mutations in histone-modifying genes have been observed in 89% of lesions [34].

### 2.2. Histone Modifications

The fundamental structural units of chromatin, the protein–DNA complex, are nucleosomes, consisting of negatively charged DNA wrapped around histones, which are positively charged proteins. Histones can undergo posttranslational modifications (PTMs) such as methylation, acetylation, phosphorylation, ubiquitination, sumoylation, and ADP-ribosylation, each of which plays a role in gene regulation and other mechanisms such as DNA repair, cell division, and cell differentiation [35]. PTMs of histones play a significant role in the pathogenesis of tumors, contributing to their proliferation and development [36]. Most types of invasive BLCA exhibit at least one histone modification [31].

Proteomic analysis of major histones in BLCA cell line 5637 revealed that PTMs vary in frequency at different histone sites. Some histone sites exclusively displayed acetylation without methylation, while others exhibited only monomethylation without acetylation. Additionally, it was found that certain histone regions simultaneously exhibited multiple distinct PTMs [37].

Among all histones, methylation of histone H3 has been extensively studied. This histone modification is of great interest because of its ability to influence gene expression, either activating genes (e.g., H3K4me3 and H3K36me3) or repressing them (e.g., H3K27me3 and H3K9me3) [36]. These modifications are mediated by mutations in the *EZH2* gene (an oncogene encoding a histone methyltransferase, also known as *HMT*) that trimethylates H3K27, leading to the silencing of the tumor-suppressor gene E-cadherin [38]. Furthermore, the modification of H3 protein is associated with a high degree of malignancy in invasive cancer. Chromatin-remodeling gene mutations are more frequently observed in BLCA than in any other cancer type (*ARID1A*, *KDM6A*, *CHD6*, or *MLL*) [35]. 

Different types of histone PTMs have varying effects on carcinogenesis. For instance, the PTM H3K4me1 correlates with advanced stages of MIBC and unfavorable survival outcomes [36]. H3K4me3 induces DNA replication and repair defects, increases chromatin instability, and serves as a potential biomarker for high-grade BLCA, as well as luminal papillary and basal squamous subtypes of BLCA. PTM H3K4me1/2 causes DNA replication and cell-cycle disturbances, induces cell invasion and migration, and is associated with recurrence and therapy resistance. PTM H3K9me2 promotes the development of NMIBC and possesses immunomodulatory effects. PTM H3K9me3 and H3K36me3 contribute to the development of higher-grade BLCA. PTM H3K27me2/3 promotes tumor development and progression in BLCA and predicts progression and recurrence in NMIBC. PTM H3K27me3 induces immune evasion of tumors, activates pro-inflammatory pathways, and leads to dysregulation of gene expression associated with cellular identity [36]. 

Various patterns of histone methylation have been identified in the context of MBIC and NMIBC, indicating the potential for distinct responses to epigenetic therapy among patients in these subgroups. Consequently, there is a pressing need to evaluate PTMs of histones for predicting responses to such epigenetic modulators. Investigation has revealed that the methyltransferase G9a is overexpressed in BLCA, and its inhibition has been demonstrated to impact cell survival through the regulation of the AMPK-mTOR pathway. Additionally, it has been reported that the interaction between G9a and DNA methyltransferase 1 (DNMT1) leads to the suppression of transcription of target genes. Furthermore, this enzyme can interact with *EZH2*, causing genetic inactivation of specific genes and, thereby, presenting a potential target for the treatment of progressive metastatic BLCA [36]. 

The diverse profiles of histone modifications are believed to reflect subtypes of luminal and basal cells. Specifically, one cluster, H3K4me1, was exclusively found in basal tumors, suggesting that the (in)activation of enhancers may contribute to the differences observed between subtypes of MIBC [39].

DNA methylation and PTMs, which dictate gene expression regulation, are reversible, making them potential targets for therapeutic intervention [40]. Inhibiting enzymes can reverse gene silencing and restore the expression of tumor-suppressor genes [31]. For instance, low concentrations of nZnO promote late-stage apoptosis of BLCA T24 cells and suppress cell invasion and migration. These antitumor effects may be attributed to elevated levels of RUNX3 because of reduced occupancy of H3K27me3 at the *RUNX3* promoter, as well as decreased levels of histone methyltransferase *EZH2* and H3K27 trimethylation [41]. 

Another potential mechanism to influence histone PTMs is through the epigenetic regulation of small noncoding RNAs (sncRNAs), such as microRNAs (miRNAs), which have an impact on histone modification regulation. In some cases, sncRNAs can target components of the histone-modification machinery, including histone methyltransferases and demethylases, thereby influencing the levels and patterns of histone modifications. This interaction between sncRNAs and histone modifications represents a regulatory mechanism that can modulate gene expression and cellular processes, including those associated with cancer development and progression [36]. Epigenetic alterations play a crucial role in the pathogenesis and progression of BLCA. Indeed, BLCA arises from multiple epigenetic events that inactivate known tumor suppressors through DNA methylation and histone modification [42].

## 3. The Role of MicroRNAs in Cancerogenesis

MicroRNAs (miRNAs) are a family of short, 18- to 25-nucleotide, noncoding RNAs that regulate gene expression at the posttranscriptional level by acting on the 3′-nontranslated mRNA regions in a complementary way [43]. It results in mRNA degradation and inhibition of protein product translation. The ability of microRNAs to inhibit translation of oncogenes and tumor suppressors suggests their involvement in carcinogenesis. Among them, there are genes involved in cell differentiation, apoptosis, and cell proliferation. Therefore, microRNAs can influence the process of tumor initiation and development. According to the microRNA database (miRbase.org), about 2000 pre-miRNAs and more than 2600 mature human microRNAs have been described to date. Current studies suggest that microRNAs can serve as biomarkers in urological cancers, including BLCA [44]. MicroRNA genes are evolutionarily conserved and distributed throughout the human genome. In some parts of the genome, they are organized into clusters. A cluster is a group of two or more microRNAs that are transcribed from physically adjacent miRNA genes, are transcribed in the same orientation, and are not separated by a transcription unit or a microRNA in the opposite orientation. A small fraction of miRNAs (about 10%) is located in the introns of coding genes. About half of all microRNAs are found within or adjacent to chromosome fragility sites, in sites of heterozygosity loss, or in regions of genome amplification in tumors [45]. 

MicroRNAs likely originated from duplication events, allowing replication of identical or similar microRNAs. MicroRNAs that have a very similar sequence and secondary structure are considered members of the same family. Because of these similarities, microRNAs in the same family often have overlapping targets, which allows for more sustained suppression of target pathways. For example, the miR-17~92 cluster is a polycistronic microRNA that consists of six microRNAs: miR-17, -18, -19a, -20, -19b, and -92. This cluster is overexpressed in various types of cancer and is involved in the development of neoplasms [46]. MicroRNA genes of the same family can be located on the same or on different chromosomes. For example, the microRNA-23b, microRNA-27b, and microRNA-24-1 genes are part of an intronic cluster in the chromosome 9q22.32 region. The six microRNA members (microRNA-30 a, microRNA-30b, microRNA-30c-1, microRNA-30c-2, microRNA-30d, and microRNA-30e) are included in the microRNA-30 family and are encoded by genes placed on chromosomes 1, 6, and 8 [47].

Transcription of microRNA genes is usually performed by RNA polymerase II with a primary transcript formation, pri-microRNA, about 1000 nucleotides in length. A required criterion is the occurrence of a self-complementary site able to provide a hairpin on the transcribe RNA. The pre-microRNA structure is identified in the nucleus and is divided from the other part of the transcript by an enzymatic complex inclusive of Drosha proteins (members of the RNase III family) and Pasha proteins (from Drosha’s partner, DGCR8). This leads to a microRNA precursor development, pre-microRNA, that is a hairpin about 60–70 nucleotides in length and represents a hairpin with a protruding dinucleotide 3′-ON end. Further maturation occurs in the cytoplasm. The pre-miRNA is transported from the nucleus by the Exp5/Ran GTP complex, where RNase III Dicer cuts off the terminal loop with the help of other RNA-binding proteins. The resulting double-stranded RNA product, about 22 nucleotides long, binds to the Ago2 protein of the Argonaute family. Subsequently, only one of the microRNA chains bound to the protein remains functional, while the second is degraded. The choice of the leading chain is determined by the structure of the duplex itself: the probability of remaining in the complex with Ago2 is greater for a chain carrying an unpaired site at its 5′ end. The mature microRNA and Ago2 protein form the so-called RISC (miRNA-induced silencing complex), which provides the main function of microRNA—suppression of gene expression. The choice of the target gene is determined by the complementarity of the microRNA seed sequence and the mRNA sequence (microRNA binding sites), which are most often located in the 3′-untranslated region [48].

## 4. Secretory microRNAs in Body Fluids

The majority of studies directed at researching microRNA expression profiles are conducted on tissue samples or cell lines (Table 1). A series of studies have demonstrated the applicability of microRNAs from biological fluids for some diseases’ diagnostics and prognosis. The use of secretory microRNAs as potential cancer biomarkers has several advantages, including their high stability in body fluids and potential availability for semi-invasive and noninvasive diagnosis. Recently, microRNAs have been stably detected in plasma and serum, where they show resistance to endogenous ribonucleases. A great deal of current information has been saved to date on circulating microRNAs and their possible use as biomarkers in different types of cancer [49]. Apart from serum and blood-plasma microRNAs, the most frequently employed for secretory biomarker analysis, microRNAs can be identified in other biological fluids. Weber, with colleagues, examined microRNA distribution in 12 human biological fluids (plasma, tears, cerebrospinal fluid, saliva, urine, breast milk, amniotic fluid, colostrum, bronchial lavage, pleural fluid, peritoneal fluid, and seminal fluid) in the healthy individuals’ group. MicroRNAs were identified in all fluids studied [50]. In general, however, urine, cerebrospinal fluid, and pleural fluid had the lowest levels of detectable microRNAs. The lowest level of detectable microRNAs in urine suggests that a significant amount of circulating microRNA is retained by the kidneys through an unknown process or is destroyed in the urine. 

To date, the origin and functions of circulating microRNAs have not been fully elucidated. There is an opinion that microRNAs mediate cell communication. 

Currently, two pathways have been proposed to incorporate microRNAs into the circulation: the passive pathway and the active pathway. In the passive pathway, microRNAs are released from cells, the integrity of which is compromised by tissue damage or apoptosis. Thus, the passive pathway does not require energy; it occurs under normal conditions and does not play a major role in circulating microRNA production [65]. 

In contrast to passive microRNA release, active secretion is accomplished by microvesicles (MVs) of cellular origin. MVs are small vesicles that can be found in almost all cell types. Many types of cells form membrane vesicles that are separated by the plasma membrane—microvesicles ranging from 100 nm to 1 µm in size. In addition, cell endosomes form invaginations that result in intraendosomal vesicle formation. If such an endosome connects to the plasma membrane and fuses with it, then the vesicles contained in it—exosomes—are released into the extracellular space. First, scientists considered such vesicles as the “trash cans” of the cell, but in 1996, Raposo et al. discovered that immune-cell-derived exosomes could function as activators of the immune system [66]. 

Later, it became evident that exosomes derived from certain cell types contain functional proteins involved in biological events. These results led to the understanding that exosomes are universal tools for intercellular communication. It was then discovered that exosomes carry nucleic acids, and in 2007, Valadi et al. discovered that exosomes contain microRNAs [67]. Nowadays, it is well-established that microRNAs in exosomes are transported between cells and suppress the expression of target genes in recipient cells [68].

Moreover, exosome secretion from cancerous tissue is much higher than from normal tissue, and higher exosomal miRNA concentrations are commonly found in tumor fluid biopsy samples such as plasma, urine, and ascites [69,70]. Together with the stability of microRNAs, this increased burden on circulating exosomes in malignancy has led to the identification of several potential biomarkers based on exosomal miRNA. 

To date, the molecular basis for microRNA stability remains unclear. However, several hypotheses exist. Circulating microRNAs packaged in MV and associated with RNA-binding proteins or chemical modifications can be protected. For example, the Argonaut proteins Ago1 and Ago2, members of the microRNA-induced gene-silencing complex, can bind and protect circulating microRNAs from degradation. Endogenous microRNAs can be transported to recipient cells through the bloodstream, involving high-density lipoprotein (HDL) [71]. Two groups of researchers also indicate that Argonaute2 (AGO2) is a key component of the plasma microRNA–protein complex responsible for the stability of unassociated microRNA vesicles. Unprotected “naked” microRNAs are more susceptible to degradation [72,73].

The active pathway of microRNA secretion is ATP dependent. First, microRNAs are transported into small secretory vesicles inside the cell and, then, enter the bloodstream, which accompanies microvesicle secretion. Thus, urinary exosomes are transported into the urinary space, with fusion of the outer membrane and the apical plasma membrane of renal tubule epithelial cells. Exosomes can originate from any type of bladder-facing epithelial cells or from the podocytes of the transitional epithelium of the bladder.

As for urine, it does not contain any unique microRNA types, and expression profile signatures observed in urine samples in patients with various types of urothelial tumors have a high value and utility as cancer biomarkers [50].

In addition, urine samples contain lower protein levels than blood samples, which reduces possible interference from protein during RNA isolation. However, they contain more nucleases, including RNAases, which lead to degradation of long chains of mRNA, which are unstable under given conditions. Unlike mRNAs, microRNAs are more resistant to degradation by nucleases because of their small size and other factors. This greater stability contributes to the reliability of analyses, with secretory miRNA being more reliable compared to miRNAs obtained from tissue samples through transurethral resection of the bladder tumor, during which miRNAs are often damaged [74].

## 5. Urine miRNAs as BLCA Biomarkers

One of the most effective and modern high-throughput technologies for searching for miRNA expression signatures is next-generation sequencing (NGS). We have found only a few studies of urinary microRNA expression analysis in BLCA using NGS analysis (Table 2). The enhanced stability of secretory miRNA compared to miRNAs extracted from tissue samples via transurethral resection of bladder tumors significantly enhances the reliability of the analyses. In the process of transurethral resection, miRNAs extracted from tissue samples frequently suffer from damage, whereas secretory miRNAs remain intact and offer more dependable results. This heightened stability of secretory miRNA underscores its superiority in ensuring accurate and trustworthy analytical outcomes when compared to miRNAs obtained through transurethral resection of bladder tumors.

In 2018, Barbara Pardini et al. performed NGS sequencing of urine samples obtained from 66 BLCA patients and 48 controls. They found differentially expressed miRNAs (miR-30a-5p, let-7c-5p, and miR-486-5p) in MIBC and NMIBC, which demonstrated high accuracy with the ability to distinguish between cases and controls (AUC model = 0.70; *p*-value = 0.01) [75]. Differential miRNA expression depending on invasiveness and degree was confirmed using qPCR on 112 cases and 65 control individuals (with 46 cases and 16 controls being an independent group of subjects, and the rest of the samples were replicas). The authors of the work came to the conclusion that noninvasive detection of the mentioned miRNAs in the urine may lead to more accurate and earlier cancer diagnosis and patient stratification [75]. Braicu et al. (2019) performed miRNA microarrays using paired tumor and adjacent normal bladder tissues with further validation using qRT-PCR of the selected transcripts. Additional next-generation sequencing investigation established the interconnection among the altered miRNAs and mutated genes. Based on the overlapping between TCGA data and data obtained in the study, they focused on the systematic identification of altered miRNAs and mutated genes involved in BLCA tumorigenesis and progression and identified 18 downregulated and 187 miRNAs upregulated. As a result of qRT-PCR validation, they selected a panel of two downregulated (miR-139-5p and miR-143-5p) and three upregulated miRNAs (miR-141b, miR-200s, and miR-205). Altered miRNA patterns are interrelated to bladder tumorigenesis, allowing them to be used for the development of novel diagnostic and prognostic biomarkers. 

Jen-Tai Lin et al. (2021) identified 50 microRNAs in urine that were differentially expressed in BLCA in comparison with controls, including 44 microRNAs with increased level of expression and 6 microRNAs with decreased level of expression, using 10 BLCA patients and 10 healthy controls. Further examination demonstrated a noteworthy upregulation of let-7b-5p, miR-149-5p, miR-146a-5p, miR-193a-5p, and miR-423-5p in BLCA tissues when compared to the corresponding adjacent normal tissues. Additionally, it was revealed that elevated expression levels of miR-149-5p and miR-193a-5p correlated significantly with unfavorable overall survival outcomes among BLCA patients. Through the implementation of qRT-PCR methodology, it was discovered that the expression levels of let-7b-5p, miR-149-5p, miR-146a-5p, and miR-423-5p were substantially increased in the urine samples of BLCA patients compared to the control group. It is important to note that one of serious limitations of this study is a small sample size; hence, it must be validated in larger cohorts in future [78]. 

Tudor Moisoiu et al. (2022) combined microRNAs and surface-enhanced Raman spectroscopy profiling of urine [77]. A retrospective cohort (BLCA = 66 and control group (CTRL) = 50) and RT-qPCR were used to confirm the selected differently expressed miRNAs. Three differentially expressed miRNAs (miR-34a-5p, miR-205-3p, and miR-210-3p) yielded an AUC of 0.92 ± 0.06 in discriminating between BLCA and CTRL, an accuracy that was superior either to miRNAs (AUC = 0.84 ± 0.03) or Raman spectroscopy data (AUC = 0.84 ± 0.05) individually. When evaluating the classification accuracy for luminal and basal BC, the combination of miRNAs and Raman spectroscopy profiling averaged an AUC of 0.95 ± 0.03 across the three machine learning algorithms, again better than miRNA (AUC = 0.89 ± 0.04) or Raman spectroscopy (AUC = 0.92 ± 0.05) individually, although Raman spectroscopy alone performed better in terms of classification accuracy. The authors came to the conclusion that miRNA profiling synergizes with Raman spectroscopy profiling for point-of-care diagnostic and molecular stratification of BLCA. 

It is important to note that the only miRNA from all the mentioned studies that overlapped in two studies is miR-205-5p. miR-205 is known to be a dual-nature miRNA, which can act either as an oncogene or as an oncosuppressor. Its role in particular cancer depends on its expression status. miR-205 actively participates in the development of numerous cancers and exerts its tumor-suppressor effects by orchestrating the regulation of multiple genes and pathways. Its established role in cancer migration and invasion involves targeting epithelial–mesenchymal transition (EMT) markers such as E-cadherin and N-cadherin. Furthermore, miR-205 directly targets *ZEB1* and *ZEB2*, further augmenting its regulatory influence. By modulating the AKT and VEGFA signaling pathways, miR-205 governs crucial cellular processes such as proliferation, cell cycle regulation, and apoptosis across various cancer types. Given its multifaceted nature, miR-205 emerges as a potent diagnostic and prognostic biomarker, capable of targeting diverse pathways and genes in cancer [79].

## 6. Role of Long Noncoding RNAs (lncRNAs) in BLCA

Long noncoding RNAs (lncRNAs) are a class of RNAs that contain more than 200 nucleotides and are unable to code proteins or peptides [80,81,82]. LncRNAs regulate gene expression and function at transcriptional, translational, and posttranslational levels and are localized predominantly in the nucleus and less frequently in the cytoplasm [83]. It has been determined that lncRNAs play an important role in the regulation of gene expression at the epigenetic, transcriptional, and posttranscriptional levels of gene expression [84]. This class of noncoding RNAs can perform both RNA-like and protein-like functions [85].

There are five main categories of lncRNAs that differ in their genomic localization. These categories are antisense lncRNAs, bidirectional lncRNAs, intronic lncRNAs, enhancer-associated lncRNAs, and intergenic lncRNAs. Intergenic and enhancer-associated lncRNAs are distinct from protein-coding genes, as they have their own promoters. Bidirectional lncRNAs are transcribed from a shared promoter and are located on the opposite strand relative to the protein-coding gene. Intronic lncRNAs are transcribed within the intronic region of a protein-coding gene [80].

LncRNAs play a crucial role in the regulation of gene expression through various mechanisms, including chromatin modification and remodeling, histone modification, and alterations in nucleosome positioning [80]. The primary mode of action for lncRNAs is the regulation of target gene expression through cis-regulation or trans-regulation [83]. Cis-regulation refers to the control of neighboring protein-coding genes by lncRNAs, while trans-regulation involves the regulation of gene expression on different chromosomes [81]. Multiple potential mechanisms have been identified through which a specific locus of lncRNA can locally modulate chromatin structure or gene expression. The lncRNA transcript itself plays a role in regulating the expression of nearby genes by attracting regulatory factors to the locus and/or modulating their activity. Additionally, the process of lncRNA transcription and/or splicing contributes to the regulatory function of the gene, independent of the specific sequence of the RNA transcript. In the case of cis-regulation, the regulatory effects solely depend on DNA elements within the lncRNA promoter or gene locus and are completely independent of the encoded RNA or its products [82,86]. 

In view of the studies showing the relationship between changes in lncRNA profiles and the pathogenesis of various diseases, including BLCA, LncRNAs are of interest as a noninvasive diagnostic marker of malignant neoplasms, especially in the context of BLCA [87,88,89,90,91]. Their broad involvement in tumor pathogenesis, encompassing such crucial processes as cell proliferation, migration, invasion, epithelial–mesenchymal transition, apoptosis, and drug resistance, highlights their importance [92,93,94,95,96]. Dysregulation of lncRNAs, either downregulation or upregulation, has been linked to BLCA initiation, proliferation, and metastasis [97]. Therefore, a comprehensive understanding of the functions of lncRNAs and their impact on cancer progression is crucial [97]. Using lncRNAs as diagnostic markers offers a less invasive approach that can potentially improve the detection of BLCA compared to traditional cystoscopy [98,99].

According to the expression pattern in BLCA, lncRNA can be upregulated and downregulated (Table 3).

### 6.1. Contribution to BLCA Carcinogenesis 

According to the findings presented in Table 3, increased expression of certain lncRNAs has been associated with the development of BLCA. These lncRNAs have been shown to influence signaling pathways involved in the progression of cancer. For instance, the lncRNA LINC00346 promotes tumor progression by regulating the PI3K/AKT signaling pathway, leading to enhanced cell proliferation, migration, invasion, apoptosis, and metastasis [149]. Urothelial carcinoma association 1 (UCA1) affects the proliferation and invasive characteristics of bladder carcinoma cells through the PI3-K-dependent pathway and under the influence of CREB [97,105]. The plasmacytoma variant 1 translocation gene (PVT1) has been observed to either inhibit or promote a malignant phenotype and WNT/β-catenin signaling in BLCA cells, although the specific molecular mechanisms remain unclear [119]. The lncRNAs transcript-1 PCAT1 and growth arrest-specific 5 (GAS5) play important roles in stimulating cancer cell proliferation and apoptosis by modulating the Wnt/β-catenin signaling pathway. Metastasis-associated lung adenocarcinoma transcript 1 (MALAT1) suppresses the migration ability of BLCA cells by downregulating EMT-related genes, such as ZEB1 and ZEB2, and the Wnt-signaling pathway while upregulating E-cadherin. Additionally, it acts as a significant mediator of TGF-β-induced EMTc [150]. Overexpression of MALAT1 leads to uncontrolled cell proliferation, migration, and clonogenicity in BLCA and can actively contribute to the initiation of cancer metastasis [102,103].

In BLCA carcinogenesis, lncRNAs can also target miRNAs. LncRNAs contain miRNA response elements (MREs) that can bind to miRNAs and prevent them from interacting with their target messenger RNAs (mRNAs), resulting in increased expression of miRNA target genes [151]. One example is homeobox transcript antisense RNA (HOTAIR), whose expression correlates with miRNA-205 and urothelial BLCA prognosis [152,153]. HOTAIR controls signaling pathways that switch from tumor suppressor to tumorigenesis [109,110]. Nuclear-enriched-abundant transcript 1 (NEAT1) affects BLCA tumor growth by participating in cell proliferation and migration through its interaction with miR-101 and miR-410 [113]. LncRNA H19, which is normally expressed during embryonic and fetal development, exhibits bidirectional regulation and can either promote or suppress tumor formation [98]. In the context of BLCA, H19 acts as an oncogene and supports cell proliferation by regulating miR-29b-3p [121,122]. Another lncRNA, small nucleolar RNA host gene 16 (SNHG16), shows increased activity in TGF-β-induced cells and BLCA tissues, and its upregulation significantly enhances bladder cell migration and invasion by affecting miRNA-150-5p expression [154]. The lncRNA terminal differentiation-inducible noncoding RNA (TINCR) is associated with an increased risk of BLCA and may regulate cell proliferation and apoptosis by influencing miR-7 expression [155]. Maternal-expressed gene 3 (MEG3), expressed in many normal tissues, plays a role in various cellular functions. MEG3 expression is reduced in cancer, and its effects on miR-494, miR-96, TPM1, miR-27a, and miR-93 have been established. Decreased expression of MEG3 induces autophagy and suppresses apoptosis, highlighting its importance in tumor suppression in BLCA [156,157,158].

The understanding of the impact of certain lncRNAs on BLCA pathogenesis is still limited. Overexpression of the second chromosomal locus associated with prostate-1 (Schlap1) was observed in BLCA tissues compared to normal bladder tissues. Silencing Schlap1 using siRNA resulted in cell growth inhibition, apoptosis induction, and migration inhibition in BLCA cells [118]. However, the specific role of Schlap1 in the pathogenesis of BLCA remains unclear in these studies. Antisense noncoding RNAs at the INK4 locus (ANRIL) are elevated in BLCA tissues compared to neighboring nontumor tissues. Silencing ANRIL using siRNA or short hairpin RNA in BLCA cells led to suppressed cell proliferation and increased apoptosis. In vivo experiments confirmed that ANRIL knockdown suppressed tumorigenicity in mice injected with EJ cells [120]. The expression levels of the lncRNA CDKN1A antisense DNA damage-activated RNA (PANDAR) were examined in urothelial BLCA patients and various tumor cell lines using real-time qPCR. Suppression of PANDAR expression through siRNA transfection resulted in decreased cell proliferation and migration, as well as increased apoptosis. PANDAR expression was significantly increased in BLCA tissues and positively correlated with higher histological grade and advanced TNM stage [128]. The role of the light ubiquitin-like modifier 1 pseudogene 3 (SUMO1P3) in BLCA pathogenesis is not well-understood. SUMO1P3 levels were significantly elevated in BLCA tissues compared to adjacent nontumor tissues. Increased SUMO1P3 expression correlated with higher histological grade and advanced TNM stage. Knockdown of SUMO1P3 using siRNA inhibited cell proliferation and migration and induced apoptosis in BLCA cells [129]. Mitochondrial lncRNAs, including sense transcript (SncmtRNA) and antisense transcript (ASncmtRNA), play crucial roles in regulating metabolism and cellular homeostasis in normal proliferating human cells. In BLCA, SncmtRNA was found to be highly expressed, while ASncmtRNA was decreased. Analysis of urine samples from BLCA patients showed increased levels of SncmtRNA but downregulation of ASncmtRNA [130].

The understanding of the effects of certain lncRNAs on BLCA pathogenesis is limited, as these RNAs have been more extensively studied in other types of cancer. One such lncRNA is gastric carcinoma highly expressed transcript 1 (GHET1). In a study involving 80 tissue samples from BLCA patients, GHET1 was found to be elevated in BLCA tissues compared to adjacent healthy tissues. Its overexpression correlated with tumor size, extent, lymph node status, and poor survival. Knockdown of GHET1 suppressed tumor cell proliferation and invasion in vitro and reversed the epithelial–mesenchymal transition in BLCA cell lines [124]. High GHET1 expression was also associated with low sensitivity to the chemotherapy drug gemcitabine in BLCA patients, and it was highly expressed in gemcitabine-resistant BLCA cell lines. Another example is colorectal-cancer-associated transcript 2 (CCAT2). In a study involving 48 BLCA patients, CCAT2 was strongly expressed in both tissues and tumor cell lines and promoted the development of BLCA cells [159]. The specific mechanistic effects of GHET1 and CCAT2 in BLCA are still not well-understood, as they have been primarily studied in other cancer types.

### 6.2. Harnessing LncRNAs as Predictive Biomarkers

Additionally, lncRNAs can exist in exosomes and are attractive for use in diagnostics and prognostics. However, their application has some limitations, including the analysis limited to known lncRNAs, false positives from nonspecific hybridizations, high variability of low-expressed genes, and rare variants of lncRNA sequences. Nevertheless, the provision of exosomal lncRNAs seems to be a promising cancer biomarker [160].

Immunity-related lncRNAs (IRlncRNAs) can be utilized for predicting immunotherapeutic responses in BLCA. Eight IRlncRNAs (MIR181A2HG, AC114730.3, LINC00892, PTPRD-AS1, LINC01013, MRPL23-AS1, LINC01395, and AC002454.1) from the Cancer Genome Atlas (TCGA) database were associated with recurrence-free survival (RFS) in BLCA using LASSO and multivariate Cox regression analysis. IRlncRNAs were also correlated with immune cell infiltration, including M0, M2, Tregs macrophages, CD8 T cells, and neutrophils, as well as biomarkers associated with immunotherapy checkpoint inhibitors (ICIs) [131]. Hypoxia-inducible factor 1α-antisense RNA 2 (HIF1A-AS2) is overexpressed in BLCA cells and tumors after cisplatin therapy. Induction of HIF1A-AS2 in BLCA cells promotes immunity against acid-induced apoptosis, and disabling HIF1A-AS2 restores sensitivity to Cys-induced apoptosis. HIF1A-AS2 inhibits the apoptotic pathway dependent on p53 family proteins by suppressing their transcriptional activity, thereby promoting the expression of highly mobile group A1 (HMGA1). Through the regulation of Wnt signaling, UCA1 increases chemoresistance in BLCA cells [117].

Several lncRNAs have the potential to serve as targets for BLCA treatment. The expression of MDC1-AS (MDC1 antisense transcript), a mediator DNA-damage checkpoint protein 1, is reduced in BLCA [161]. An RT-qPCR study involving 80 samples from gastric cancer patients revealed that MDC1-AS inhibits gastric oncogenesis through an MDC1-dependent mechanism. Knockdown of MDC1 attenuated the inhibitory effect of MDC1-AS on MKN28 cell proliferation and metastasis, while MDC1 overexpression mitigated the stimulatory effect of MDC1-AS knockdown [162]. Analysis of data from 41 pairs of BLCA tissues and adjacent normal bladder tissues demonstrated reduced levels of MDC1-AS and MDC1 expression in BLCA. After MDC1-AS overexpression, elevated levels of MDC1 were observed in BLCA cells. Additionally, an inhibitory role of MDC1-AS in the behavior of malignant BLCA cells was discovered [139]. A cohort study of human BLCA tissue samples, with benign controls for LOC572558 expression, revealed a significant reduction in LOC572558 expression in BLCA cell lines and tissues. Moreover, ectopic expression of LOC572558 suppressed cell proliferation and motility, induced S-phase cell cycle arrest, promoted cell apoptosis, and was associated with dephosphorylation of AKT and MDM2, as well as phosphorylation of the p53 protein. Therefore, it is evident that LOC572558 acts as a tumor suppressor and regulates the p53 signaling pathway in BLCA [136]. The expression of “deleted” in the chromosomal region of BLCA candidate 1 (DBCCR1) is suppressed by promoter hypermethylation in 50% of the analyzed BLCA cell lines. Exogenous expression of the DBCCR1 protein or epitope-labeled HA fusion protein, HA-DBCCR1, in NIH3T3 cells and human bladder tumor cell lines results in the suppression of proliferation. Cell-cycle analysis in NIH3T3 cells demonstrated that DBCCR1-mediated growth suppression was associated with an increase in the number of cells in the G(1) phase of the cell cycle [163].

As mentioned earlier, several in vitro studies have demonstrated the efficacy of lncRNA knockdown techniques in inducing apoptosis or inhibiting cell proliferation and migration in BLCA cells. It is worth emphasizing that several lncRNAs have demonstrated different roles, including oncogenic and tumor suppressor roles, in different cancers. This may be attributed to the relative importance of certain signaling pathways in each cancer type or the more tissue-specific signatures of lncRNAs compared to protein-coding genes. Consequently, the role of each lncRNA must be evaluated in different biological contexts. Such expression analysis will pave the way for identifying processes involved in BLCA initiation and/or progression, such as increased invasion and proliferation of cancer cells, as well as factors contributing to treatment resistance.

## 7. Role of Circular RNAs (circRNAs) in BLCA Carcinogenesis

Circular RNAs (circRNAs) are a class of endogenous noncoding RNAs in eukaryotic cells. In contrast to the structure of coding RNAs, circRNAs lack a 5′-cap and a 3′-polyadenylated tail [164]. Due to this structure, they are protected from degradation by RNA exonucleases [165]. Due to the development of NGS-sequencing technologies, a large number of expressed circRNAs have been identified in normal and malignant human tumor cells. CircRNAs are specifically expressed in a number of organisms and at different stages of disease [166]. The nucleotide sequence of circRNAs is evolutionarily conserved, indicating that they are stable and selective [167]. The role of circRNAs has also been confirmed in gene expression and the pathogenesis of a number of disorders. The functions of circRNAs mediate modulation of gene expression at different levels, and depending on the gene site, circRNAs are subdivided into exonic circRNAs, intronic circRNAs, or exon-intronic circRNAs [168]. The presence of circRNA indicates its cell- and tissue-specific properties; when circRNA levels are elevated in certain diseases, they can be considered as potential diagnostic biomarkers in terms of therapeutic potential, which is particularly important in malignant neoplasms [169]. Studies have shown that circRNAs regulate the proliferation, invasion, migration, and apoptosis of multiple malignant cell types [170]. Multiple circRNAs have been detected to be differentially expressed in BLCA tissues and to play a huge potential role in the study of BLCA pathogenesis. Using microarray circRNA data from four paired BLCA tissues and adjacent normal bladder tissues, Zhong et al. determined 3243 and 469 circRNAs that were differentially expressed in BLCA tissues compared with normal tissues, of which 285 were markedly increased and 184 were reduced [171]. One example of a circRNA involved in carcinogenesis in BLCA is the circRNA_0058063 in tumor tissues of patients with BLCA compared to normal tissues [172] (Table 4). 

In BLCA cells, downregulation of circ_0058063 suppressed cell proliferation and tumor cell migration and enhanced apoptosis processes by directly inhibiting the expression of miR-145 targeting the CDK6 gene [184]. Further studies showed that disabling circRNA_0058063 blocked cell proliferation and cell invasion and promoted cell death, suggesting that circRNA_0058063 may modulate BLCA progression. Thus, circRNA_0058063 has some potential as a prognostic biomarker and therapeutic target for pathology detection and treatment [170]. A study by Young et al. demonstrated that circUVRAG levels were significantly increased in BLCA tissues and cell lines, and its knockdown sharply inhibited cell proliferation by promoting miR-223, which led to suppression of the FGFR2 [185]. CircUVRAG expression has been shown to be significantly elevated in tumor cells, and inhibition of circUVRAG represses BLCA cell proliferation and migration. In vivo studies have shown that suppression of circUVRAG regulation resulted in a significant reduction in the volume and weight of tumor xenografts compared to controls [186]. 

One study by Su et al. demonstrated that circRIP2 enhances BLCA progression through the Tgf-2/smad3 signaling pathway by inhibiting miR-1305 [175]. This confirms that dysregulation of the circRIP2/smad3-mediated Tgf-2/smad3 signaling pathway is also involved in BLCA progression. In addition to regulating the tumor itself, circRIP2 may have some effect on the tumor microenvironment, ultimately leading to suppression of tumor growth. To confirm this hypothesis, the RNA sequencing of circRIP2 overexpression was tested. In addition to Tgf-β signaling, chemokines (CCL2, CCL3, CXCL5, CXCL17, and CXCL20) and cytokines (IL-6, IL-13, and IL-17) that particularly interact with immune cells were also found to change significantly. These data suggest that the oncogenic circRIP2 may play a tumor-suppressive role by interacting significantly with the microenvironment [175].

Some types of circRNAs exhibit anti-oncogenic effects. Studies suggest that ciRS-7 can play a role as an oncogenic circRNA by binding to miR-7, leading to the progression of a number of malignancies, including lung cancer [187], stomach cancer [178], colorectal cancer [177], and esophageal cancer [176]. However, in BLCA, Cdr1as plays a completely different role. In a study of a group of mice overexpressing Cdr1as, tumor volume and weight were significantly reduced when BLCA cell lines infected with adenovirus-expressing Cdr1as were injected subcutaneously, which is consistent with the in vitro results. Cdr1as has also been shown to perform anti-oncogenic functions by binding to miR-135a [188].

In the vast majority of studies, the dysregulation of circRNA expression correlated with the unfavorable clinicopathological characteristics of patients. For example, hsa_circRNA_403658 was previously stated to be elevated in BLCAa tissues compared to neighboring adjacent tissues. High expression levels of hsa_circRNA_403658 correlate with shorter survival in patients with BLCAa, and evaluation of the association between hsa_circRNA expression and clinicopathologic characteristics revealed a positive correlation between the severity of hsa_circRNA_403658 expression and such [173] characteristics as larger tumor size (≥3 cm vs. <3 cm), distant metastasis, and late TNM stage (III-IV) [173]. Another experiment found that downregulation of hsa_circ_0077837 and circ_0004826 in BLCA was linked to deterioration of overall survival and recurrence-free survival in BLCA patients. Studies have confirmed that both circRNAs can act as independent prognostic markers in BLCA [179]. Table 4 also presents some types of circRNAs whose up- or downregulation has been noted in a number of studies and was a predictor of negative/positive prognosis in BLCA. 

Due to significant advances in RNAi research, circRNAs have attracted attention from researchers and are gradually becoming a new trend in the study of malignancies. Due to their stability, high prevalence, and presence in various body fluids, they may be prognostic biomarkers, as well as potential treatment targets for BLCA.

## 8. Immune Components in BLCA

### 8.1. Immune Biomarkers of BLCA

Despite the large number of genetic and epigenetic biomarkers promising for prognosis diagnosis and BLCAs, immunological features and markers of the immune response have been underemphasized, despite, for example, immune checkpoint inhibitors that are well-known in cancer therapy. It has already been approved (particularly, PD1/PD-L1 and CTLA4 inhibitors) in several cancers and is now estimated to be in the context for urogenital cancers, including the BLCA [189]. On the immunological side, it is also worth noting the possibility of using inflammatory biomarkers.

The relationship between different types of inflammatory markers and prognosis in BLCA has been described in detail in the publications [190]. Predictive relevance of serum content of pro-inflammatory mediators such as IL-6, CRP, and TNFά has been demonstrated in some types of malignancies [191,192]. CD8+ T-cell content as a marker was assessed by flow cytometry in one study [193]. The quantitative content of CD8+ T cells in peripheral blood was found to be inversely related to tumor infiltration by CD8+ T cells (r2 = 0.63, *p* < 0.0001). In a multivariate analysis, a low blood CD8+ T-cell count was associated with fewer intravesical recurrences after TURB (HR = 0.4, 95% CI 0.17–0.94). A high infiltration by cytotoxic T lymphocytes (CTL) is typically correlated with a positive prognosis in BLCA [160,194] but is also associated with a high risk of recurrence in NMIBC [195]. The presence of considerable infiltration of macrophages and regulatory T cells (Tregs) is generally associated with tumor progression [196], but elevated Tregs levels have also been shown to be associated with a positive prognosis in CD [197]. One of the fundamental areas of study of the immune component in malignant neoplasms and specific markers is the study of the cellular microenvironment of the tumor.

### 8.2. Cell Heterogeneity in BLCA

The microenvironment of the tumor is crucial for the progression of the neoplasm, metastasis, and recurrence and significantly influences treatment outcomes [198].

A groundbreaking study was conducted on specimens from the Cancer Genome Atlas (TCGA) to provide the first comprehensive and quantitative characterization of bladder cancer (BLCA). A total of 131 BLCA samples, all of which had no prior exposure to chemotherapy, underwent full-genome sequencing and RNA sequencing. This comprehensive analysis aimed to identify somatic mutations and copy number aberrations and transcriptomic subtypes associated with BLCA. Most of the 32 mutated genes are identified as significant and involved in cell-cycle regulation, modifications of chromatin structure, and kinase-signaling pathways [199].

In another study that used single-cell transcriptome sequencing technology, an atlas of the cells of the entire BLCA microenvironment was compiled. At the sequencing of single-cell RNA-based droplet (scRNA-seq) from eight bladder carcinoma tumor samples and three tumor samples (three para-tumor samples), 19 clusters of tumor, immune, and stromal cells with important functions were identified, indicating high intratumor heterogeneity. The authors revealed that bladder carcinoma cells express the lowest levels of MHC-II molecules. In the same way, a subset of LAMP3+ dendritic cells also expressed diverse cytokines, such as CCL17, CCL19, and CCL22, facilitating the immunosuppressive TME. Two types of cancer-associated fibroblasts (CAFs) have been identified that accelerate tumor progression [86].

Various solid tumors are characterized by the presence of immune cells such as T- and B-lymphocytes, natural killer (NK) cells, macrophages, antigen-presenting cells, neutrophils, and dendritic cells. Because of abnormal differentiation of altered immune cells, they demonstrate diverse behavior and morphology [200].

These abnormal deviations in differentiation can sometimes be stimulated by epigenetically controlled lineage-specific changes that affect gene expression. All of these are crucial for maintaining the identity of immune cells and ensuring cellular responses to various external and internal stimuli [201,202]. DNA methylation modification is associated with a multitude of biological functions in cancer, such as tumor microenvironment formation and evolution, as well as restoration of immune-cycle disruptions [203,204]. DNA methylation can contribute to changes in the tumor microenvironment (TME); for instance, DNA hypomethylation alters the expression of genes such as CTLA-4, TIGIT, and PD-1 [205]. Combining PD-1 agents with demethylation inhibitors may represent a more effective immunotherapeutic strategy compared to current approaches [206].

As an example, the investigations carried out by some authors have demonstrated that different patterns in the methylation process are involved in the activation of myeloid and lymphoid cancer cells. It was found that reverse methylation patterns were observed in myeloid and lymphoid cancer tissues, with lymphoid neoplasms lacking CpG methylation patterns, while myeloid malignant neoplasms have significantly increased DNA methylation levels [207].

In naive CD4-positive T cells, the interferon-γ gene promoter and enhancer are methylated. Nevertheless, in human type 1 lymphocytes (Th1), in which IFN-γ expression is induced, the IFN-γ gene promoter and enhancer are demethylated, implicating an importance in Th1/Th2 differentiation [208].

One of the most prominent immune components of tumor tissue is tumor-associated macrophages (TAMs). Depending on the degree of polarization, the M1 and M2 macrophage subpopulations contribute differently to tumor progression [209]. M2 macrophages are prooncogenic, and their marker genes are epigenetically controlled by cross-methylation changes in histone H3 lysine-4 and histone H3 lysine-27. IL-4 was observed to also induce a decrease in H3K27me2/3 levels on the promoter domains of the M2 marker genes. They were partially reestablished upon Jmjd3 knockout, which demonstrates that Jmjd3 is actively implicated in the dynamic regulation of H3K27me2/3. In macrophages treated with IL-4 and knocked down by Jmjd3 was found upregulation of M2 marker genes. Knockdown of Jmjd3 induced a remarkable reduction in M2-specific marker genes but not a complete one. This might be the result of the possibility that other functional pathways are engaged in the regulation of M2-specific marker genes [210].

It is well-known that NK cells are a type of cytotoxic lymphocytes that belong to a rapidly expanding family of known innate lymphoid cells that are part of the innate immune system to control tumor growth [211]. Some evidence suggests that histone acetylation is responsible for the coordination of NK cell activation and their effector roles [212]. The vast majority of NK cells express the NKG2D receptor; it plays an essential role in tumor-cell recognition. NKG2D is a type 2 transmembrane protein and is coupled in the human membrane to the signal adaptor molecule DAP10 [213]. NKG2D associates with each of the multiple major histocompatibility complex-I-like ligands encoded by the host genome, including *MICA*, *MICB*, and *ULBP1-6* in humans [214]. In the variety of normal cells, NKG2D ligands are lowly expressed, but one or more are commonly overexpressed on the surface of a majority of cancer cells, and it is worth nothing that expression of NKG2D ligands on the surface of cells enhances their receptiveness to removal by NK cells [215].

NKG2D-deprived knockout mice demonstrate a stronger incidence or burden of cancer in multiple cancer models, and the role of NKG2D in cancer immunosurveillance, including genetically engineered models of spontaneous cancer [216].

The cell plasticity is the ability of cells to have various phenotypes [217]. Cellular plasticity in malignant tumors brings additional conceptual complexity of tumor heterogeneity because plasticity is reciprocal and can develop either through dedifferentiation or transdifferentiation [218]. In BLCA, cellular plasticity was observed by Yang et al. with the use of single-cell sequencing approaches, revealing that non-stem cells in urothelial malignant tumors can take on a stem-like origin and become capable of self-refreshing [219].

A study by Warrick et al. discovered that diverse regions of the same tumor often demonstrate various molecular variations. They used the classification system of Lund University, Sweden, to classify 83 histologically variable bladder tumors into multiple molecular subsets. In the cohort of their study, they reported that 39% of malignancies with multiple types of histologies expressed molecular heterogeneity [220]. Urothelial carcinoma is the most frequently detected histologic form among all malignant tumors of the urinary system [221]. Bladder malignancy is characterized by high cellular heterogeneity, which is characterized by different histological, molecular and clinical phenotypes [222,223,224], which complicates tumor differentiation. 

Tumor cells are stimulated by metabolites, stromal cells, and signaling molecules. This allows tumor cells to constantly switch from a differentiated state to a cancer stem cell state. This allows tumor cells to constantly switch from a differentiated state to a cancer stem cell state [217,218,219,220,221,222,223,224,225]. It also enables them to adapt to the constantly changing microenvironment, avoid immune surveillance, and metastasize [226]. The microenvironment of a tumor is an interconnecting structure involving tumor cells, stromal tissue (immune cells, fibroblasts, cytokines, and vascular tissue) as a whole, and the surrounding extracellular matrix [227].

New gene markers were also found in some clusters. For example, *PLA2G2A* and *IGFBP6* were highly expressed in fibroblast 1 compared with those in other clusters. *PIGR* and *LCN2* were also highly expressed in umbrella cells.

In the study, the researchers discovered two novel human bladder cell types: ADRA2A+ and HRH2+ interstitial cells and TNNT1+ epithelial cells, which are also present in the bladder tissue of both rats and mice. Studies have discovered that urothelial cells give rise to the largest number of bladder tumors [228].

Overall, the findings showed that the intratumor and intertumor heterogeneity of MIUBCs demands the development of personalized interventions against internal mechanisms of tumor cells and complicated interactions of cell subpopulations [229,230,231,232,233]. Consequently, it is necessary to observe not only the interactions between cell subpopulations but also the involvement of the immunological components of BLCA regulated by epigenetic modifications [234,235,236,237,238].

## 9. Conclusions

With the considerable progress in RNA research, it has attracted the increased attention of researchers and has gradually become a new frontier in cancer research.

A huge number of epigenetic regulatory mechanisms, microRNA transcripts, circRNAs, and lncRNAs have been detected, and some of them have proven to be functional ncRNAs related to malignant phenotypes and clinical manifestations. In this review, we not only presented the epigenetic mechanisms and features of noncoding RNAs, but also the role of immune system components as clinical biomarkers in the diagnosis and prognosis of BLCA and discussed the roles and significance of these biomarkers.

Our results suggest that some intrinsic and extrinsic features of tumor cells are associated with clinical outcomes in BLCA. If supported by more research, these could potentially assist in guiding future treatment

## Figures and Tables

**Table 1 ijms-24-09854-t001:** MicroRNAs, involved in regulation of cancer cell proliferation.

miRNA	Target	Up-(↑) or Down-(↓) Regulation	Experimental Models (Cell Lines, Animal Models)	References
miR-221	TRAIL	↑	Cell lines	[51]
miR-129	SOX4, GALNT1	↓	Normal and tumor bladder samples	[52]
miR-1/133a/218	LASP1	↓	Tissue samples	[53]
miR-19a	PTEN	↓	Cell lines	[54]
miRs-30a-3p	KRT7	↓	Cell lines, normal and tumor bladder samples	[55]
miR-34a	CDK6	↓	Cell lines and tissue samples	[56]
miR-99a/100	FGFR3	↓	Normal and tumor bladder samples	[57]
miR-125b	E2F3/cyclin A2	↓	Clinical BLCA and normal bladder tissue. BLCA tissues and cell lines	[58,59]
miR-145/133a	FSCN1	↓	Cell lines	[60]
miR-145	CBFB, PP3CA	↓	BLCA tissues and cell lines	[61]
miR-200s/205	ZEB1, ZEB2	↓	Cell lines	[62]
miR-200 family	ERRFI-1/EMT	↓		
miR-203	Bcl-w	↓	BLCA tissues	[63]
miR-517a	Bclaf1	↓	Cell lines	[64]

**Table 2 ijms-24-09854-t002:** Diagnostic urine microRNAs detected by NGS.

Investigation	Sample Size	Upregulated miRNA	Downregulated miRNA	Diagnostic Accuracy (Overall AUC)	References
Pardini et al., 2018		let-7c-5p, miR-30a-5p	miR-486-5p	0.70 (for panel of 3 miRNA)	[75]
Braicu et al., 2019		miR-141-3p miR-205-5p miR-200b-3p	miR-139-5p miR-143-5p	0.86 (miR-141-3p) 0.89 (miR-205-5p)	[76]
Lin et al., 2021	10 malignant BLCA tissues and 10 normal BLCA tissues	Let-7b-5p miR-146a-5p miR-149-5p miR-193a-5p miR-423-5p	-		[75]
Moisoiu et al., 2022	15 BLCA patients and 16 controls	miR-34a-5p miR-205-5p miR-210-3p		0.84 (for panel of 3 miRNA)	[77]

**Table 3 ijms-24-09854-t003:** LncRNAs involved in regulation of cancer cell proliferation.

lncRNA	Target	Up-(↑) or Down-(↓) Regulation	Experimental Models (Cell Lines, Animal Models)	References
LINC00346	PI3K/AKT	↑	human BLCA tissue, BLCA cell lines	[100,101]
MALAT1	ZEB, ZEB2, WNT	↑	human BLCA tissue, mice model,	[102,103,104]
UCA1	PI3-K, Wnt	↑	human BLCA tissue	[97,105,106]
PCAT1	EMT and Wnt/β-catenin	↑	human	[107,108]
HOTAIR	microRNA-205	↑	human BLCA tissue, BLCA cell lines, murine model	[109,110,111,112]
TUG1	miR-320a/FOXQ1	↑	human BLCA tissue, BLCA cell lines	[113,114]
NEAT1	miR-101/VEGF-C, miR-410	↑	human BLCA tissue, BLCA cell lines	[115,116]
HIF1A-AS2	p53 family proteins	↑	human BLCA tissue, BLCA cell lines	[117]
Schlap1	N/A for BLCA	↑	BLCA cell lines	[118]
PVT1	WNT/β-catenin	↑	human BLCA tissue, BLCA cell lines	[119]
ANRIL	N/A for BLCA	↑	human BLCA tissue, BLCA cell lines	[120]
H19	miR-29b-3p	↑	human BLCA tissue, BLCA cell lines	[121,122,123]
GHET1	N/A for BLCA	↑	human BLCA tissue	[124]
CCAT2	N/A for BLCA	↑	human BLCA tissue	[125]
SNHG16	miRNA-150-5p, p21	↑	human BLCA tissue, BLCA cell lines	[123,126]
TINCR	miR-7	↑	human BLCA tissue, BLCA cell lines	[127]
PANDAR	N/A for BLCA	↑	human BLCA tissue	[128]
SUMO1P3	N/A for BLCA	↑	human BLCA tissue	[129]
SncmtRNA	N/A for BLCA	↑	human	[130]
IRlncRNA	N/A for BLCA	↑	human BLCA tissue	[131]
RBAT1	E2F3	↑	human BLCA tissue, BLCA cell lines	[132]
Linc00312	miR-197-3p	↓	human BLCA tissue	[101,133]
BANCR	MAPK	↓	human BLCA tissue	[134,135]
Loc572558	AKT, MDM2	↓	human BLCA tissue	[136]
MIR31HG	N/A for BLCA	↓	human BLCA tissue	[137,138]
MDC1-AS	MDC1	↓	BLCA cell lines	[139]
DBCCR1	DNMT1	↓	BLCA cell line	[95,140]
MEG3	miR-494, miR-96, TPM1, miR-27a, miR-93	↓	human BLCA tissue, BLCA cell lines	[141,142,143,144]
GAS5	Wnt/β-catenin, CDK6, EZH2	↓	human BLCA tissue, BLCA cell lines	[145,146,147,148]

**Table 4 ijms-24-09854-t004:** CircRNAs involved in regulation of cancer cell proliferation.

miRNA	Target	Up-(↑) or Down-(↓) Regulation	Experimental Models (Cell Lines, Animal Models)	References
circRNA_0058063	CDK6 gene	↓	cell lines	[173]
circUVRAG	FGFR2	↓	BLCA tissues and cell lines	[174]
circRIP2	Tgf-2/smad3	↑	BLCA tissues and cell lines	[175]
ciRS-7	miR-7	↓	BLCA tissues	[176,177,178,179]
hsa_circRNA_403658	HIF-1α	↑	BLCA tissues and cell lines	[173]
circ_0077837	no data	↓	BLCA tissues and cell lines	[179]
hsa_circ_0004826	no data	↓	BLCA tissues and cell lines	[179]
circRGNEF	KIF2C	↑	animal models	[180]
circ_0068871	FGFR/STAT3	↑	BLCA tissues and cell lines	[181]
CircTFRC	TFRC	↑	BLCA tissues, cell lines, and animal models	[182]
CircFOXO3	TGFBR2	↓	BLCA tissues and cell lines	[174]
CircNR3C1	Cyclin D1	↓	BLCA tissues and cell lines	[183]

## Data Availability

Not applicable.

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
