# Peer review of "Epigenetic and Immunological Features of Bladder Cancer"

_ijms, 2023, doi:10.3390/ijms24129854_

Round 1

Reviewer 1 Report

The manuscript provides a comprehensive overview of epigenetic regulation, especially in the realm of microRNAs, lnRNAs and circular RNAs and bladder cancer, biomarker candidates are included along with considerations of the immune system. While comprehensive a few sections lacked focus and was a bit difficult to transition from one topic to the next. Suggestions and concerns are listed below:

1. Sections on DNA methylation and histone acetylation are less detailed than the sections pertaining to the RNAs. There was also much more detail proved about the molecular machinery associated with RNA regulation compared to DNA methylation or histone acetylation. Only including the RNA modifiers would be an appropriate refocusing of the initial discussion.

 2. On Page 4 the section entitled “Role of miRNA in bladder cancer”, while comprehensive in explaining miRNAs this section could be bolstered with additional discussion of the literature in terms of which miRNAs have been associated with different cases or with suppression of specific genes associated with bladder cancer, however this level of detail is provided in subsequent sections. Perhaps retitling this section to be background information on miRNA etc , but this section does not go into the specifics with bl cancer yet so would benefit from retitling

3.    General comment on references, often the reference is proved at the end of the paragraph, but it is difficult if that reference captures all of the statements in the several sentences prior or only the last sentence, placing the reference at the end of the sentence for which it specifically refers to would be beneficial to readers.

Minor

1.Section 2 at the top starts out with a description of the section content that should be deleted

2.Section 2.1 should not lead off with the word "Thus."

3.If including DNA methylation, more detail is needed on the molecular events, similar to the level of detail provided for the RNA sections. Page 3 Lines 120-123 would fit better in the beginning of section 2.1 as they describe what DNA methylation is and how it is acquired.

4. Page 5 line 239 this sentence does not make sense and the data leading to observing urine miRNAs in patients with different urothelial tumors should be explained in greater detail.

5. Page 7 references are needed for several of the statements especially the section on miR-205.

6. Grammar should be checked throughout the manuscript

English is appropriate however grammar should be checked throughout the manuscript.

Author Response

Dear reviewer!

We are so grateful for your valuable comments and suggestions for our article. We appreciate the attention shown to our work and have been in awe of correcting our article. Thanks to this we were able to delve even deeper into the subject matter of the article and hopefully present our data more correctly.

The sections on DNA methylation and histone acetylation have now been described in more detail. We have described the chapter on DNA methylation and histone acetylation in more detail.

On page 4, the section 'Role of miRNAs in bladder cancer', has been renamed.

We have rearranged the links according to your recommendations/

Minor comments have also been fixed.
Added text is marked in yellow. 

Reviewer 2 Report

Irina et al. summarized the research on epigenetic and immunological features of bladder cancer. This work provides a comprehensive review of recent advances. The manuscript is well-organized and I recommend its publication after some revisions.

1. This article is entitled “Epigenetic and Immunological Features” of bladder cancer. However, in this work, the content of epigenetic alteration and immunological pattern was quite imbalanced, and no close connection between these two parts was presented in this article. I would recommend deletion of the immune part to make this article more focused. Or the part of immune should be revised to focus on how the immunological features of BC were regulated by epigenetic modifications.

2. The content of reference literature has been described too in detail (especially “6. Role of long non-coding RNAs (lncRNAs) in BLCA”). In my opinion, this content should be integrated and more concise.

3. This article provides abundant supporting literature to present the epigenetic alterations of BC. But the prognostic value of these alterations is not emphasized. Additional parts focusing on the association of epigenetic features and prognosis and treatment response would be helpful (such as doi: 10.1080/07357907.2022.2146703; doi: 10.3390/cancers14040866.)

Minor:

1. please delete the template content on Page 2 Lines 64-66.

2. Page 3 Lines 120-123 should be moved to Page 2 Lines 67?

Author Response

Dear reviewer!

We are so grateful for your valuable comments and suggestions for our article. We appreciate the attention shown to our work and have been in awe of correcting our article. Thanks to this we were able to delve even deeper into the subject matter of the article and hopefully present our data more correctly.

1.The immunological part has been deeply revised and now focuses on the relationship between immunological features of BLCA and epigenetic changes

2.  The role of long non-coding RNAs (lncRNAs) in BLCA") is fragmented and significantly shortened

3.The prognostic value of epigenetic changes in BLCA has been added to the article. Additional parts on the association of epigenetic features with prognosis and response to treatment have also been included.

All minor comments have been resolved, and all added text is highlighted as yellow.

Reviewer 3 Report

The manuscript entitled “epigenetic and immunological features of bladder cancer” by Gilyazova et al. aims to summarize the most significant advancements in the epigenetics of BC. Overall, the paper deals with an interesting and contemporary topic which is widely discussed in the literature. To this regards, the manuscript could further enrich the current panorama and report the most recent findings. Few corrections are required in order to improve the quality of the work and increase the readability and the clarity of it. The suggested corrections are reported followingly.

INTRODUCTION

22-33: Add further data regarding the epidemiology of BC as well as the known risk factors. To this regard, please see DOI: 10.2144/fsoa-2020-0210 and doi: 10.3390/jcm12020728

Considering that this is a narrative review, it is not needed to report the search strategy – otherwise, you should transform your work into a systematic review.

EPIGENETIC ALTERATION OF BLCA

64-66: Please check typos. Include a brief introduction before proceeding to describe subparagraphs.

ROLE OF LONG NON-CODING RNAs

This paragraph is too long and should be divided into smaller subparagraphs.

IMMUNE COMPONENTS IN BLCA

To this regard please also see: doi: 10.3390/ijms23031133

CONCLUSIONS

A paragraph reporting future perspectives as well as limitations of the current panorama regarding the topic discussed should be reported.

check typos and grammar along the text.

Author Response

Dear reviewer!

We are so grateful for your valuable comments and suggestions for our article. We appreciate the attention shown to our work and have been in awe of correcting our article. Thanks to this we were able to delve even deeper into the subject matter of the article and hopefully present our data more correctly.

22-33: Additional data have been added concerning the epidemiology of BLCA as well as known risk factors. 

EPIGENETIC VARIATION BLCA

64-66: All typos have been eliminated.  A brief introduction has been included before we move on to describe the sub-items.

THE ROLE OF LONG NON-CODING RNC

This paragraph has been divided into smaller sub-paragraphs.

IMMUNE COMPONENTS IN THE BLCA

Suggested citations have been added in all marked paragraphs

CONCLUSIONS

A paragraph reporting future perspectives has been included, also the limitations of the current situation in relation to the topic under discussion.

All minor comments have been rectified, with all added text highlighted in yellow

Round 2

Reviewer 2 Report

The authors have revised the manuscript according to my suggestions and solved my questions.